# Enzymes in the Cholesterol Synthesis Pathway: Interactomics in the Cancer Context

**DOI:** 10.3390/biomedicines9080895

**Published:** 2021-07-26

**Authors:** Pavel Ershov, Leonid Kaluzhskiy, Yuri Mezentsev, Evgeniy Yablokov, Oksana Gnedenko, Alexis Ivanov

**Affiliations:** Institute of Biomedical Chemistry, 10 Building 8, Pogodinskaya Street, 119121 Moscow, Russia; la-kaluzhskiy@yandex.ru (L.K.); yu.mezentsev@gmail.com (Y.M.); evgeyablokov1988@mail.ru (E.Y.); gnedenko.oksana@gmail.com (O.G.); professor-ivanov@yandex.ru (A.I.)

**Keywords:** cholesterol, cholesterol precursors, pathway, enzymes, protein partners, interactome, cancer, tumor, TCGA, protein–protein interaction

## Abstract

A global protein interactome ensures the maintenance of regulatory, signaling and structural processes in cells, but at the same time, aberrations in the repertoire of protein–protein interactions usually cause a disease onset. Many metabolic enzymes catalyze multistage transformation of cholesterol precursors in the cholesterol biosynthesis pathway. Cancer-associated deregulation of these enzymes through various molecular mechanisms results in pathological cholesterol accumulation (its precursors) which can be disease risk factors. This work is aimed at systematization and bioinformatic analysis of the available interactomics data on seventeen enzymes in the cholesterol pathway, encoded by *HMGCR, MVK, PMVK, MVD, FDPS, FDFT1, SQLE, LSS, DHCR24, CYP51A1, TM7SF2, MSMO1, NSDHL, HSD17B7, EBP, SC5D, DHCR7* genes. The spectrum of 165 unique and 21 common protein partners that physically interact with target enzymes was selected from several interatomic resources. Among them there were 47 modifying proteins from different protein kinases/phosphatases and ubiquitin-protein ligases/deubiquitinases families. A literature search, enrichment and gene co-expression analysis showed that about a quarter of the identified protein partners was associated with cancer hallmarks and over-represented in cancer pathways. Our results allow to update the current fundamental view on protein–protein interactions and regulatory aspects of the cholesterol synthesis enzymes and annotate of their sub-interactomes in term of possible involvement in cancers that will contribute to prioritization of protein targets for future drug development.

## 1. Introduction

In normal cells, cholesterol is a crucial component of biomembranes. It plays a regulatory and signaling role interacting with cholesterol-binding proteins. Cholesterol is the precursor of steroid hormones and bile acid biosynthesis pathways [1]. In the last ten years, cholesterol emerged as a modulator of tumor progression [2,3,4,5,6]. Cholesterol accumulation is a characteristic feature for cancer cells [5,7]. Cholesterol-containing membrane raft domains regulate certain types of proteins, including various cell signaling ones that are critical to tumor cell survival and invasiveness. This suggests that membrane rafts have a regulatory role in tumor progression [8]. Exogenous cholesterol intake was positively associated with higher incidence of stomach, kidney, pancreas, lung, breast, bladder, colon, rectum cancers and lymphoma [7].

Enzymes of cholesterol synthesis pathway represent molecular targets for drugs, mainly, statins. Although the number of registered clinical trials of statin therapy in cancer treatment has reached about two hundred (keywords: disease—cancer; other terms—statin; (https://clinicaltrials.gov/, accessed on 1 June 2021), statins efficacy remains quite controversial due to their pleiotropic effects [9,10]. Statin application was associated with reduction of cancer risk by 20% [11]. However, the addition of statins to standard anticancer therapy does not improve overall survival [12,13].

Involvement of enzymes of cholesterol synthesis pathway (cholesterol synthesis enzymes) in provoking and maintaining pathological cancer-dependent processes is closely linked with regulation on transcriptional and post-translational levels [1,14]. The last one can be realized through protein–protein interactions (PPIs) and post-translational modifications (PTMs) resulting in a change of protein stability and enzymatic activity.

With the growing number of experimentally determined PPIs of cholesterol synthesis enzymes, it is necessary to assess and make comparative analysis of available information on both interactomes of individual enzymes (sub-interactomes) and the group interactome for several enzymes, functioning in a single metabolic pathway. The aim of the work was to systematize interactomics data on the cholesterol synthesis enzymes with respect to the cancer context. This is intended to update the current understanding of the functional and regulatory aspects of the cholesterol synthesis pathway in socially significant diseases.

## 2. Materials and Methods

### 2.1. TCGA Datasets Analysis

The Cancer Genome Atlas (TCGA) program has generated, analyzed, and made available data on the genomic sequence, expression, methylation, and copy number variation of over 11,000 individuals with over 30 different types of cancer [15,16]. The TCGA database was used to determine the expression patterns of genes, encoding cholesterol synthesis enzymes, as well as to find the associations with patient survival. The Gene Expression Profiling Interactive Analysis (GEPIA2) web server [17] (http://gepia2.cancer-pku.cn/, accessed on 23 April 2021) was used to obtain median values of gene expression in tumor and normal tissues from TCGA datasets and to determine the tumor/normal ratio or fold change (FC). The selection of differentially expressed genes (DEGs) was performed according to FC ≥ 3 and cut-off significance level 0.01. A heat map was plotted using web server NG-CHM GUI 2.19.1 [18]. Prognostic value of DEGs and Kaplan–Meier plotting were performed using GEPIA2 with the following settings: significance level 0.05; p-value adjustment by FDR; group cut-off-median. Principal component analysis (PCA) of data matrix with FC values for 17 genes, encoding cholesterol synthesis enzymes, and 30 different cancers was performed using ClustVis resource [19] (https://biit.cs.ut.ee/clustvis/, accessed on 26 April 2021). The assessment of mutational variability (somatic mutation frequency) of genes was performed using TCGA PanCancer Atlas Studies dataset (10953 patients/10967 samples) at the cBioPortal (https://www.cbioportal.org/, accessed on 2 May 2021).

### 2.2. Interactomics Data Acquisition and Processing

All possible spectrum of experimentally established (physical interactions) protein partners for each of 17 cholesterol synthesis enzymes (target enzymes) was retrieved with a set of interactome browsers [20,21,22,23,24,25,26,27,28,29,30,31,32] (Table 1). Common protein partners for target enzymes were obtained using the Venn diagram tool (http://bioinformatics.psb.ugent.be/webtools/Venn/, accessed on 26 March 2021).

### 2.3. Annotation and Enrichment Analysis

Functional annotation of the final list of target enzyme protein partners with Gene Ontology, KEGG, REACTOME, Panther (FDR ≤ 0.05) and Wiki Pathways terms (*p*-value ≤ 0.05) was performed with over-representation analysis (ORA) on the Web Gestalt server (WEB-based Gene SeT AnaLysis Toolkit) [33]. Annotation of proteins by subcellular localization was performed using NextProt database (https://www.nextprot.org/, accessed on 29 March 2021). Gene annotation by “common essential genes” category was done using CRISPR-Cas9 whole-genome drop out screens to identify dependencies in cancer cells [34] on the website Cancer Dependency Map (https://score.depmap.sanger.ac.uk/, accessed on 2 April 2021) and DepMap portal (https://depmap.org/, accessed on 2 April 2021). Pubmed (https://pubmed.ncbi.nlm.nih.gov/, accessed on 10 June 2021) and Litsense [35] (https://www.ncbi.nlm.nih.gov/research/litsense, accessed on 10 June 2021) resources were used to search for available literature data. Annotation of cancer driver genes was carried out using Cancer Gene Census portal (https://cancer.sanger.ac.uk/census, accessed on 5 April)).

Prediction of post-translational modification (PTM) sites of cholesterol synthesis enzymes was performed using the web-server PhosphoSitePlus (https://www.phosphosite.org/, accessed on 4 May 2021) containing experimental results of mass-spectrometry PTMs mapping.

## 3. Results

### 3.1. The Main Characteristics of Cholesterol Synthesis Enzymes

Seventeen enzymes (target enzymes), functioning in a single metabolic cascade of multi-step transformations of cholesterol precursors (Figure 1), were selected from a pathway module M00100 KEGG (https://www.genome.jp/kegg-bin/show_pathway?map00100, accessed on 3 March 2021) and Hallmark Cholesterol Homeostasis (ID M5892, 74 genes, Molecular Signature Database (http://www.gsea-msigdb.org/, accessed on 6 March 2021)). The main characteristics of the cholesterol synthesis enzymes are also shown in Appendix A.

Approximately 70% of cholesterol synthesis enzymes represent membrane proteins containing one or several transmembrane domains localized, mostly, in the endoplasmic reticulum membrane (ERM), Golgi apparatus membrane, cytoplasmic vesicles, plasma membrane and nucleus inner membrane (Appendix A). The remaining 30% of enzymes (MVK, PMVK, MVD, FDPS and LSS) are localized in the cytosol, peroxisomes and lipid droplets. Different subcellular localizations of enzymes may also suggest the additional functions besides transformation of cholesterol precursors. Multiple functions for this group of enzymes are poorly studied, possibly due to insufficient data on the functional significance of mapped PPIs. For example, the MoonProt database [36] (http://www.moonlightingproteins.org) does not contain information about multiple functions of the target enzymes, except the mevalonate kinase (MVK). It is known that lanosterol synthase (LSS) can regulate protein aggregation by effectively reducing the number and/or size of sequestosomes/aggresomes formed by endogenous proteins in the normal and cancer cells [37]. In addition, farnesyl pyrophosphate synthase (FDPS) obtains an additional function in fibroblast growth factor (FGF) signaling pathway through binding to the FGF-receptors [38]. It is interesting to note that delta(24)-sterol reductase (DHCR24) can protect neuronal cells from ER stress-induced apoptosis by attenuating ER stress signaling, possibly through scavenging intracellular reactive oxygen species and elevating cholesterol levels [39].

The oligomeric state of target enzymes is an important interactomic factor. It was shown that a dimeric form is characteristic for cholestenol delta-isomerase (EBP) [40], sterol 14-alpha-demethylase (CYP51A1) [41] and MVK [42]. Squalene epoxidase (SQLE) exists in a monomeric form [43], while 3-hydroxy-3-methylglutaryl-CoA reductase (HMGCR) forms tight tetramers that implies the influence of the enzyme’s oligomeric state on the activity and mechanisms for allosteric modulation by ligands [44]. Since half of these data concerns non-human orthologues, the concept of the oligomeric state of most human cholesterol synthesis enzymes is not yet completely studied.

Many cholesterol synthesis enzymes for catalytic reactions require coenzymes NADH and NADPH, which intracellular levels can be correlated with tumor progression under condition of metabolism reprogramming in cancers [45]. At present, no coenzyme-dependent oxidoreductases, which can bind with target enzymes, have been identified [46], except for CYP51A1 belonging to the multigene cytochrome P450 family. It strongly requires electron transfer from its direct redox partner NADPH-dependent cytochrome P450 oxidoreductase (POR) for catalysis [47]. It is interesting to note that SQLE contributes to cancers via its metabolites. An increase in SQLE expression promotes the cholesteryl ester production to induce hepatocellular carcinoma cell growth. Moreover, SQLE can raise the NADP+/NADPH ratio what triggers DNA methyltransferase 3A (DNMT3A) expression, DNMT3A-mediated epigenetic silencing of phosphatase tensin homolog (PTEN) and activation of pro-oncogenic mammalian target of rapamycin pathway [48]. Thus, levels of intracellular coenzymes and their ratio can be important factors in proper functioning of enzymes in the cholesterol pathway.

Then, we searched for literature data on involvement of 17 target enzymes in carcinogenesis and important findings are shown in Table 2.

Thus, the associations between functioning, at least, three quarters of the cholesterol synthesis enzymes and solid cancers progression were found [48,49,50,51,52,53,54,55,56,57,58,59,60,61,62,63,64,65,66,67,68,69,70,71,72,73,74,75,76,77,78,79,80,81,82,83,84,85,86,87,88,89,90,91]. Not only overexpression and post-translational activation of enzymes correlated with pathogenic abnormalities, but also a decrease in catalytic function due to down-regulation of expression or pharmacological inhibition (SC5D, NSDHL and EBP). Another important implication is that several target enzymes have differentiated contributions to cancers depending on the tissue specificity.

### 3.2. Transcriptomic Landscape of Genes, Encoding Cholesterol Synthesis Enzymes

The panoramic transcriptome profile of 17 target genes is shown in Appendix A. It can be conditionally distinguished two groups of tumors. The first group includes COAD, DLBC, PAAD, READ and THYM tumors, while the second one includes SKCM, LAML, CHOL and TGCT tumors with a preferential up-regulation or down-regulation of DEGs, respectively. Principal component analysis (PCA) showed that, in fact, only the *CYP51A1* and *NSDHL* genes had the closest expression profiles in almost all tumors (Appendix A). A clusterogram, visualizing PCA output, revealed, at least, four large tumor clusters with similar profiles of DEGs (Figure 2): (1)—KIRC and KIRP; (2)—ACC, GBM, SKCM, HNSC, ESCA, etc.; (3)—BRCA, PAAD, BLCA, COAD, READ, etc.; (4)—LAML, DLBC, THYM, LIHC, LGG, LUAD, KIRC and LUSC. It is worth mentioning that alterations of expression profiles of target genes are often due to binding of cancer-activated master regulators in their promoter regions. It can be demonstrated, for example, by the network R-HSA-2426168 from Reactome database (https://reactome.org/, accessed on 20 April 2021)). Master regulators such as SREBF1, SREBF2, SP1 and NF-Y have broad promoter binding specificity and are subjected to modulation through several oncogenic signaling pathways [92,93,94,95,96], but this “transcription regulation issue” is out of scope of the present work.

Five genes from these maps meet the strictest selection criteria: p < 0.001, hazard ratio (HR) ≤ 0.5 or HR ≥ 2 as well as a number of cases ≥ 200. The maps of overall sur-vival (OS) and disease-free survival (RFS) for 17 target genes are shown in Appendix A respectively (Appendix A). For all of them (OS: *CYP51A1* in KIRC, *FDFT1* in KIRC, *HMGCR* in KIRC, *SC5D* in KIRC, *TM7SF2* in LGG; RFS: *FDFT1* in KIRC, *SC5D* in PRAD) the Kaplan–Meier plots are shown in Appendix A. It should be noted that multigenic signatures have a higher priority in prognostic value than single transcriptome markers [97]. First of all, survival analysis of a panel of 17 target genes versus each TCGA tumor was performed (Appendix A). It is shown that the selection criteria were met in the case of KIRC only. More negative prognosis for survival was associated with lower profile of cholesterol biosynthesis genes expression (Appendix A). It should be noted, that the severe cholesterol metabolism disrupting was observed in the KIRC cells [98,99,100]. We also carried out the survival analysis for each target enzyme versus all tumors (pan-cancer prognostic significance). Results obtained are presented in Appendix A, but no hits were found. Finally, the gene signature 1 (*HMGCR*, *DHCR7*, *SC5D*, *NSDHL*, *CYP51A1* and *FDFT1*) (Appendix A) and the gene signature 2 (*SC5D*, *TM7SF2* and *MVD*) (Appendix A) were found to have prognostic value for KIRC and LGG tumors, respectively. Thus, high expression of these genes can be associated with the decrease of mortality by 2.5 times (HR value = 0.4).

### 3.3. Interactomics Landscape of Cholesterol Synthesis Enzymes

In total, 2239 experimentally determined protein partners for 17 target enzymes were found by means of interactome browsers (Table 1). Search statistics are shown in Appendix A. Found protein partners were then selected in two consecutive rounds: by the coincidence of protein partners found by, at least, three different interactome browsers and by the similar profiles of subcellular localization of a target enzyme and its interaction partners (Figure 3). In addition, protein names and Uniprot IDs for each protein partner are listed in Appendix A. List A, containing 165 unique protein partners for all target enzymes, was used for functional annotation. List B, containing in total 186 proteins (165 unique and 21 common protein partners), was then used to visualize the PPI network or, if it may say so, cluster interactome of cholesterol synthesis enzymes (Figure 4).

Somatic mutation frequency of genes, encoding cholesterol synthesis enzymes, ranged from 0.4–0.7% and they were not annotated as cancer driver genes. Annotation of genes, encoding 165 unique protein partners of enzymes, showed that following ones were found as cancer drivers with corresponding somatic mutation frequency values: PRKACA (0.7%); EWSR1 (1.2%); NDRG1 (0.8%); FGFR1 (1.4%); IDH1 (5.1%); PABPC1 (0.9%); LMNA (0.8%); MYH9 (3.2%); SUZ12 (1.0%); TP53 (35.3%); HRAS (1.2%); IKBKB (1.4%); CYLD (1.3%); ERBB2 (2.9%). Cancer-dependent PPIs can consist of a protein encoded by a driver gene [101]. It was showed that a vast proportion of mutations in driver genes occurred in the binding interfaces of two proteins, which means that “non-canonical” protein complexes can emerge in tumor tissue, and vice versa, formation of canonical biologically significant complexes in normal tissue can be suppressed during neoplastic transformation. Due to some of the found protein partners, encoded by cancer driver genes, it was hypothesized for protein complexes with their participation to be directly regulated by carcinogenesis.

Functional annotation of found protein partners with Gene Ontology (GO) terms is shown in Appendix A. Over-representation analysis (ORA) revealed molecular functions enrichment (“transferase activity”, “chaperone binding”, “protease binding”, “ubiquitin-like protein binding”, “ubiquitinyl hydrolase activity”) (Appendix A) and biological processes enrichment (“endoplasmic reticulum to cytosol transport”, “nucleobase-containing small molecule interconversion”, “protein exit from endoplasmic reticulum”, “ER-nucleus signaling pathway”, “response to topologically incorrect protein”, “response to endoplasmic reticulum stress”) (Appendix A). ORA of protein partners in KEGG, REACTOME, Panther and WikiPathways databases showed their involvement in nucleotides biosynthesis and metabolism, amino acids pathways, synaptic signal transmission, protein processing in endoplasmic reticulum (ER), ER quality control (ERQC) and cancer-related pathways (apoptotic pathway, MAPK family signaling cascades, metabolic reprogramming and central carbon metabolism in cancer, Ras signaling, RAC1/PAK1/p38/MMP2 pathway and three cancer-specific pathways in prostate, endometrial and bladder cancers) (Appendix A). Thus, overall pool of protein partners, forming PPIs with cholesterol synthesis enzymes, can be divided into two big groups involved in protein processing and cell signaling.

Taking into account the over-representation of protein partners in cancer-related pathways, we tried to find the associations of each binary PPI with the cancer-specific context. A way to hypothesize such associations is to correlate of expression profiles of two genes encoded protein products, forming a PPI (gene co-expression analysis). A positive or a negative correlation in different tumor tissues indirectly point to such associations. Co-expression analysis was performed only for those tumor tissues, in which DEGs (tumor/normal fold change ≥ 3), encoding the cholesterol synthesis enzymes, were observed (Appendix A). A Pearson correlation coefficient (R) equal to 0.4 was used as a cut-off level. An average (|0.4 < R < 0.6|) and strong correlation (|0.6 < R < 0.7|) of gene co-expression were found in five different tumor tissues for 42 and 7 binary PPIs, respectively (Table 3). R-values were mostly positive, however, a negative correlation was observed for PPIs with participation of EBP in thymoma tissues (339 and 118 normal and tumor cases, respectively) (Table 3). The Human Proteome Atlas database (https://www.proteinatlas.org) was then used for comparison of tissue-specific expression profiles for each binary PPI with co-expressed genes. Concordance of the gene and protein expression profiles can be seen only for HMGCR─VCP, DHCR24─CKAP5 and DHCR7─FADS1 protein interactions in rectum adenocarcinoma datasets (318 and 92 normal and tumor cases, respectively) as well as for DHCR7─FADS1 in pancreatic adenocarcinoma (171 and 179 normal and tumor cases, respectively). However, it turned out that protein expression data for several cholesterol synthesis enzymes and their protein partners were absent in tumor tissues.

### 3.4. Post-Translational Modifications (PTM)

Further, we analyzed the known PTM spectrum of cholesterol synthesis enzymes as well as the spectrum of modifying proteins among found protein partners. The amino acid sequences of the target enzymes contain a large number of phosphorylation and ubiquitylation sites (Appendix A). Forty-seven (25%) protein partners which can potentially function as modifying proteins were summarized in Appendix A. It should be noted that 15 proteins (AMPK, PP2A, CTSL, RNF145, gp78, USP20, MEK5, GSK3β, MARCH6, IDOL, PKC, AMFR, Itch, PRKACA, SYVN1) were described as modifying proteins [71,102,103,104,105,106,107,108,109,110,111,112,113,114,115,116,117,118,119] (Appendix A). Results of annotation of all modifying proteins with cancer hallmarks and cancer pathways terms are shown in Figure 5.

Figure 5A shows that ubiquitin-protein ligases and protein kinases were the main functional classes of modifying protein partners which was in agreement with the known PTMs sites on the cholesterol synthesis enzymes (Appendix A). The diagrams in Figure 5B,C and the data in Appendix A demonstrated that protein kinases (AMPK, PRKACA, MEK5/ERK5, GSK3B, PKC and ILK), protein phosphatases (PP2A, PPP1CB, DUSP6 and PTPN5) and proteases (CTSL, CASP7 and CAPN1) are highly enriched in terms of cancer hallmark and cancer pathways terms.

## 4. Discussion

### 4.1. Mapping of Protein–Protein Interaction

PPIs represent four major types: (1)—two proteins form a direct complex through physical contacts; (2)—proteins interact with the same protein, without forming a direct complex with each other (indirect complex formation); (3)—both proteins are combined in a single metabolic cascade, but do not form either a direct or indirect complex; (4)—a functionally significant multi-protein complex, with each component interacting physically with one or more adjacent components [120,121]. Studies of sub-interactome of an individual protein are aimed at mapping of a repertoire of PPIs for elucidating regulatory and signaling pathways as well as annotating other unknown functions of a protein by known functions of its protein partners. Previously, we have studied sub-interactomes of clinically significant enzymes thromboxane and prostacyclin synthases, which are associated with several human pathologies, including cancer. It was found that two enzymes with 20% amino acid sequence identity had both common and tissue-specific protein partners [122,123,124]. In general, the identification of PPIs of a target protein by affinity chromatography and mass spectrometry showed that the number of its potential protein partners can reach several dozen [122,125,126,127], and the complete experimental validation of all combinations of binary PPIs is an overly complex methodological problem.

Studies of sub-interactomes and the construction of global PPIs networks allow to form comprehensive understanding of complexity of cellular processes and determine a variety of interactomic profiles which can discriminate molecular events occurring in normal and disease states [128,129,130]. Since cancer is one the most widespread disease in the human population, many scientific reports focus on identifying not only DEGs or proteins (DEPs), but also on establishing cancer-specific differential protein–protein interactions (dPPIs) [131]. Thus, one can reasonably suggest that endogenous and exogenous factors provoking carcinogenesis could alter the PPIs repertoire.

### 4.2. Protein–Protein Interactions of Cholesterol Synthesis Enzymes in the Cancer Context

#### 4.2.1. SQLE

Cancer-associated small integral membrane open reading frame 1 protein (CASIMO1) is overexpressed in breast tumors and interacts with SQLE. It is interesting that overexpression of CASIMO1 leads to SQLE protein accumulation, while knockdown of CASIMO1 decreased SQLE protein [132].

The SQLE, like HMGCR, is believed to be a proto-oncogene and marker of aggressive colorectal cancer (CRC) while interacting with GSK3B and p53. It was shown that SQLE reduction caused by cholesterol accumulation aggravates CRC progression via the activation of the β-catenin oncogenic pathway and deactivation of the p53 tumor suppressor pathway [71].

#### 4.2.2. CYP51A1

PGRMC1 (progesterone receptor membrane component 1) is required for modulation of microsomal P450 cytochromes in yeast and humans by binding with CYP51A1 and positively regulates it. Loss of PGRMC1 function reduces CYP51A1 activity and increase production of toxic sterol intermediates [133]. In addition, PGRMC1 was shown to interact with FDFT1, SCD1 (SCD5 homologue) and is overexpressed in hormone receptor-positive breast cancer that correlated with enhanced cancer cell proliferation and lipid raft formation [134].

#### 4.2.3. TM7SF2

It is known that Beclin 1 is an essential autophagy protein and has been shown to play a role in tumor suppression [135]. Interactions of Beclin 1 with TM7SF2 [136] may also have a significance in cancer-dependent autophagy [137].

#### 4.2.4. MVD

MVD enzyme was identified as a binding partner of the mortalin which belongs to the HSP70 protein family and is involved in cell senescence and immortalization pathways. There is an indication that MVD/mortalin interaction affects the activity of p21(Ras) and its downstream modulators of normal and cancer cell proliferation in the Ras-Raf pathway [138].

#### 4.2.5. DHCR24

Hepatitis C virus induced overexpression of DHCR24 enzyme in human hepatocytes was resulted in resistance to inhibition of the p53 stress response by stimulating the accumulation of the MDM2—p53 complex in the cytoplasm and inhibition of the p53 acetylation in the nucleus [139]. It follows that DHCR24 and SQLE enzymes in the cholesterol synthesis pathway can act not only as the targets for modifying enzymes but also as a molecular switches, triggering the alternative p53-dependent cancer cascades.

Thus, at least for five enzymes SQLE, FDFT1, MVD, CYP51A1 and TM7SF2, the significance of PPIs with their participation in the cancer context was shown, so the revealing of other cancer-specific PPIs through gene co-expression analysis, functional and clinical annotation of common protein partners for target enzymes can help to expand current understanding of cancer interactomics.

### 4.3. Common Protein Partners of Cholesterol Synthesis Enzymes

From Figure 4, it follows that there were several common proteins HSCB, CREB3, MOV10 and VKORC1, which interacted with three or more target enzymes in the cholesterol synthesis pathway. In fact, common proteins are of certain importance because they act as “connectors” in the PPI network and could link different enzymes’ sub-interactomes. Twice as many protein partners (ABCE1, UBL4A, PNKD, SYVN1, FAF2, SRPRB, PNKD, REEP5 and KSR1) were found to interact with only two different enzymes (Figure 4). Further, we discussed below the possible associations between PPIs with participation of common protein partners and cancer-related events.

#### 4.3.1. Molecular Chaperones

The accumulation of misfolded proteins can promote pathological processes due to dysfunction of molecular chaperones. Heat shock cognate B (HSCB) is a co-chaperone; it inserts Fe-S cluster into a number of proteins but does not have intrinsic chaperone activity and lacks a domain necessary for interaction with misfolded proteins [140]. CYP51A1, NSDHL, PMVK, MSMO1 and EBP enzymes do not have Fe-S clusters but interact with HSCB, so, it could be implied that HSCB may function as a scaffold to facilitate the positioning of protein substrates on HSCA which possesses chaperone activity [141].

ATP-binding cassette transporter (ABCE1), being a chaperone, associates with chemotherapy resistance in glioma via PI3K/Akt/NF-κB pathway [142,143] form PPIs with EBP and FDPS enzymes.

#### 4.3.2. Ubiquitin-Protein Ligases

HRD1 complex (ID 6859, Corum database, https://mips.helmholtz-muenchen.de/corum/) mediates ubiquitin-dependent degradation of misfolded proteins. It consists of 13 proteins, including AUP1, ERLIN2, FAF2, HSP90B1 and SYVN1, which form PPIs with several cholesterol synthesis enzymes (Figure 4). E3 ubiquitin-protein ligase synoviolin (SYVN1), interacting with HMGCR and FDFT1, establishes the crosstalk between endoplasmic reticulum associated degradation (ERAD) and p53-mediated apoptosis under ER stress in proliferative diseases [144,145].

Fas associated factor family member 2 (FAF2) interacts with HMGCR and SQLE (Figure 4). FAF2 is considered to be a potential therapeutic target due to its regulatory effect on the oncogenic Ras signaling via disrupting the stability of GTPase-activating protein neurofibromin [146] and an essential determinant of metabolically stimulated degradation of HMGCR [147]. It is assumed that, by analogy with HMGCR, a similar way of post-translational regulation of protein stability can take place for SQLE.

#### 4.3.3. Metabolic Enzymes

Alpha-enolase (ENO1) is a well-studied multifunctional enzyme with a large PPI network consisting of 441 interactors (BioGRID database, https://thebiogrid.org/108338). A pool of recent findings has confirmed a strong relationship between ENO1 overexpression and cancer development [148,149]. ENO1 forms PPIs with EBP and MVK enzymes, with the latter showing a positive correlation of gene expression profiles with ENO1 (Table 3). PPIs with participation of ENO1 could be regarded as an important element in the initiation of cancer-related events.

Vitamin K 2,3-epoxide reductase subunit 1 (VKORC1) is, presumably, capable to PPIs with disulfide isomerase (PDI) [150] and regulated by pro-oncogenic mTOR signaling [151]. Interesting, that VKORC1–EBP and VKORC1–HSD17B7 interactions were described in [152], but associations with cancers are still unrevealed.

#### 4.3.4. Signaling Proteins

Signal recognition particle receptor (SRPRB) plays a role in cell proliferation and apoptosis, probably, via NF-κB pathway [153]. SRPRB belongs to the Ras family of small GTPases and can take part in translocation of proteins in the ER membrane [154]. Due to the lack of evidences on exact SRPRB function, the biological meaning of HMGCR–SRPRB or FDFT1–SRPRB interactions remains unclear.

Paroxysmal non-kinesiogenic dyskinesia protein (PNKD) causes disordered cell differentiation, initiates malignant transformation and accelerates metastasis [155,156]. It forms different heterogeneous protein complexes [155], and, therefore, the interactions of FDFT1–PNKD and LSS–PNKD can be considered from regulatory point of view.

Receptor expression-enhancing protein 5 (REEP5) interacts with DHCR24 and EBP (Figure 4). REEP5, as well as REEP6, modulates oncogenic signaling through C-X-C chemokine receptor type 1 (CXCR1): the depletion of REEP5 and REEP6 significantly reduced cell growth and invasion via downregulating IL-8 mediated ERK phosphorylation, actin polymerization and the expression of genes related to metastasis [157]. Co-expression analysis showed a positive correlation of *REEP5* and *DHCR24* gene expression profiles in PRAD and BRCA tumors (Table 3).

#### 4.3.5. Transport Proteins

Annotation of SC5D–BSCL2 and EBP–BSCL2 interactions is possible through the known function of BSCL2 (seipin) in the lipid droplet (LD) biogenesis. BSCL2 supports the formation of structurally uniform contacts between ER membrane and LDs [158]. Besides, BSCL2 facilitates the delivery of molecular cargo from ER to LDs in human cells [159]. Thus, the educated guess could be made about involvement of ER membrane bound proteins SC5D and EBP into the ER-LD contacts.

#### 4.3.6. Modifying Proteins

The dysregulation of protein tyrosine phosphatases is associated with several human diseases, including cancers. PTPN1 directly dephosphorylates the terminal kinase c-Jun N-terminal kinase (JNK) of the MAPK pathway [160]. Negative regulation of the JNK/MAPK pathway by PTPN1 was found to reduce the tumor necrosis factor α (TNFα)-dependent cell death response [161]. However, it is unknown if “non-signaling” PTPN1’s protein substrates exist.

### 4.4. Regulation of Cholesterol Synthesis Enzymes through Post-Translational Modifications (PTM)

#### 4.4.1. PTMs of HMGCR Are the Most Studied

Ubiquitin-dependent proteolytic degradation of HMGCR occurs under cholesterol-replete conditions. This provokes insulin-induced gene 1 (INSIG1) to bind to the sterol-sensing domain of HMGCR and to recruit E3 ligases with subsequent ubiquitination of HMGCR [162,163]. Ubiquitination of HMGCR contributes to the recognition by the ATPase VCP/p97 complex that mediates extraction and delivery of HMGCR from ER membranes to cytosolic 26S proteasomes [164]. Theoretically, in this machinery, a cancer-driven reduction of *INSIG1* expression could dysregulate HMGCR proteolytic degradation under cholesterol-replete conditions and promote excess cholesterol accumulation. Analysis of TCGA datasets shows that, indeed, *INSIG1* expression was reduced in 6 from 19 tumors and was statistically significant only in CHOL and LUAD compared to normal tissues. However, under these conditions, no gene co-expression between *INSIG1* and *HMGCR* was observed. Additional analysis of CHOL expression datasets (GSE132305, 182 tumor and 38 normal tissue samples) using NCBI-GEO depository (https://www.ncbi.nlm.nih.gov/gds, accessed on 18 July 2021) showed that *INSIG1* logFC value was 0.32 (adj.P.Val = 0.026, Bonferroni&Hochberg correction). As for LUAD, we found logFC value 0.33 (adj.P.Val = 6.15 × 10^−5^) for *INSIG1* expression (GSE63459 consisting of 30 paired adjacent and tumor tissue samples). In another dataset GSE43458, we compared 30 samples from non-smokers vs 40 tumor samples obtained from non-smokers and from smokers. It was shown that *INSIG1* expression was characterized logFC values 0.31 (adj.P.Val = 0.01) and 0.5 (8.41 × 10^−5^), respectively. It should still be said that there was no obvious convergence between TCGA and NCBI-GEO datasets. Nevertheless, it is interesting that a positive Pearson correlation (R > 0.6) between up-regulated *INSIG1* and *HMGCR* expression in the TCGA-READ dataset (Table 3) may indicate the cancer-type specific expression of *INSIG1*.

On the other hand, interaction of HMGCR with UbiA prenyltransferase domain-containing protein-1 (UBIAD1) and heat shock protein 90 (HSP90) inhibits HMGCR proteolysis. These PPIs can be referred cancer-related molecular events under conditions of *HSP90* and *UBIAD1* overexpression which is associated with tumor progression, up-regulation of the mevalonate pathway and cholesterol accumulation [165,166].

Proteolytic stability of HMGCR may also be negatively regulated by adenosine monophosphate-activated protein kinase (AMPK) [102], with its activity being closely correlated with tumor suppression when AMPK modulates mTOR and Akt pathways [104]. De-phosphorylation of HMGCR was mediated by PP2A and PP2C phosphatases resulted in an increase of HMGCR activity [105]. PP2A was shown to suppress tumors but PP2A is often down-regulated in tumors and its re-activation can induce apoptosis. However, factors of PP2A inactivation are still unknown [106].

#### 4.4.2. A Model of PTM Regulation of Different Parts of Cholesterol Synthesis Pathway

As demonstrated above, the post-translational regulation (activity and stability) of HMGCR realized through PPIs and protein modifications (i.e., ubiquitination and phosphorylation/de-phosphorylation) have been studied most carefully. This cannot be said about PTM patterns of other enzymes, involved in the cholesterol synthesis pathway. Assuming the presence of similar PTM patterns for most of enzymes, “mechanics” of HMGCR regulation should be extrapolated to those enzymes in the metabolic pathway, which PPI spectrum was only fragmentarily characterized from the point of biological meaning. That favors, firstly, by the presence of common modifying proteins for different cholesterol synthesis enzymes (Appendix A) and, secondly, by the coincidence of the annotated functions of protein partners that interact with them. In Figure 6, we visualize the PTM spectrum and modifying proteins in the various “parts” of the cholesterol pathway: initial part (HMGCR), middle part (SQLE and CYP51A1) and distant part (DHCR24 and DHCR7). There are, at least, four common modifying proteins (AMFR, MARCH6, PRKACA and PTPN5). Each of them can potentially affect several target enzymes from different parts of the pathway. For example, MARCH6 interacts with SQLE, CYP51A1 and DHCR24 as well as AMFR interacts with HMGCR, DHCR24 and DHCR7. In view of the fact that MARCH6 and AMFR are poly-specific E3 ubiquitin-protein ligases, it could be suggested that there are conservative PPIs, participating in the proteolytic stability regulation of the cholesterol synthesis enzymes. It is interesting to note that E3 ubiquitin-protein ligases are applied in targeting low molecular weight degradators based on the Proteolysis Targeting Chimera (PROTAC technology) [167]. In brief, one “arm” of an PROTAC inhibitor specifically interacts with a target protein while the other is covalently linked to the E3 ubiquitin-protein ligase. The recruitment of the latter to a target protein makes it possible to trigger ubiquitin-dependent proteasome degradation. Thus, one of the strategies for therapeutic managing the cholesterol pathway can be linked with PROTAC technology development.

### 4.5. A Multiprotein Cholesterol “Metabolon”

Obviously, the question of whether the enzymes, functioning in the cholesterol synthesis pathway, physically interact with each other to form a multi-protein cluster (“metabolon”) in human cells, remains not entirely clear. Such spatial clustering of enzymes would create conditions for more energetically favorable and effective transformation of cholesterol precursors. In the FunCoup database (https://funcoup5.scilifelab.se/search/), human protein interactions FDFT1-MSMO1, LSS-HSD17B1, MSMO1-SQLE and MSMO1-CYP51A1 were predicted on interactions between orthologous yeast proteins Erg11p, Erg25p, Erg27p, Erg26p and Erg28p. Erg proteins form a protein–protein core anchored in the membrane that recruits other proteins in the lipid metabolism pathway [168]. Protein interactions between human DHCR24 and DHCR7 [169], in part, favor in the cholesterol “metabolone” hypothesis. On the other hand, DHCR7, DHCR24 and EBP enzymes might not always combine in a functional complex because they potentially play different roles that go beyond the cholesterol biosynthesis pathway [170]. Thus, an existence of multiprotein cholesterol “metabolon” in human cells is a subject for future investigations.

### 4.6. A landscape of Cholesterol Precursors in the Cancer Context

A repertoire of PPIs can be organically replenished with protein–ligand interactions. Here, ligands mean low molecular weight metabolites, i.e., cholesterol precursors, which should be expected to bind with different cellular proteins producing down-stream biological effects. However, this scientific area is quite poorly studied and further, we focus on the direct or indirect involvement of cholesterol precursors in cancers. We found that cholesterol precursors can be formally divided into three groups (Table 4) depending on their impact on carcinogenesis [50,171,172,173,174,175,176,177,178,179,180,181,182,183,184,185,186,187,188,189,190,191,192].

Group I includes metabolites, which can promote carcinogenesis. Squalene does not accumulate in normal cells. When squalene epoxidase is inactivated, an excess of squalene altered the cellular lipid profile and protected anaplastic large cell lymphoma from ferroptotic cell death, providing a growth advantage of tumors under oxidative stress [181]. A variety of squalene effects was discussed in more detail in the review [182].

Cancer-associated molecular mechanism with positive-feedback loop between mutp53 and the mevalonate pathway exist. Mevalonate-5-phosphate contributed to stabilization of mutp53 promoting tumor progression by up-regulation of genes, encoding the mevalonate pathway enzymes. It was reported that pharmacological inhibition of mevalonate kinase or its knockdown triggers ubiquitin-dependent degradation of mutp53 [180]. Significant mevalonate accumulation has been found in some tumors [172,173] and this metabolite can affect tumors through modulation of EMT, remodeling of the cytoskeleton, cell motility and polarity (non-canonical Wnt/planar pathway) [193].

Group II (Table 4) includes metabolites having an onco-protective role in suppressing tumor growth and proliferation. It was shown that an increase of lanosterol level caused the suppression of aberrant foci of the colon crypt and consequently reduced their total amount [183]. The anti-aggregation activity of lanosterol [37] might facilitate the cells to recycle proteins from aggregates indicating that lanosterol functions in the cancer-related proteostasis regulation. Meiosis activating sterol (MAS) 4,4-dimethyl-cholesta-8,14,24-trienol (FF-MAS) is a ligand for liver X receptor alpha (LXRα) which mediated regulation of cholesterol efflux and uptake through the transcriptional regulation of cholesterol transporters and low-density lipoproteins (LDL) via E3 ubiquitin-protein ligase inducible degrader of the LDL receptor (IDOL) [184]. In addition, IDOL was found to be a modifying protein regulating stability of squalene epoxidase [112]. MAS are substrates of NSDHL enzyme and inhibition of NSDHL was observed in tumors with activated EGFR-KRAS signaling [81]. Vitamin D precursor, 7-dehydrocholesterol, has been shown to induce apoptosis and inhibit melanoma cell proliferation as well as invasion [185]. It is interesting that vitamin D supplementation reduced the incidence of advanced cancer (hazard ratio 0.62 [95% CI, 0.45–0.86]) [187].

Metabolites from Group III did not show any significant effects in tumors (Table 4) and their metabolites levels can be fluctuated depending on external factors, e.g., chemical treatment. The inhibitory effect of phenyl acetate on the prostate cancer cells proliferation was found, leading to a decrease in farnesyl pyrophosphate accumulation. At the same time, exogenous farnesyl pyrophosphate reduces the effect of phenyl acetate on the cell proliferation and apoptosis [188]. It can be noted that inhibition of zymosterol biosynthesis occurred in melanoma cells [189] while treatment of rat hepatoma cells with 24(R,S),25-imino-lanosterol and triparanol caused accumulation of zymosterol and desmosterol [190]. Application of carcinogens 3-methylcholanthrene and 12-O-tetradecanoylphorbol-13-acetate in mouse models resulted in elevation of 7-dehydro-desmosterol and desmosterol levels, but cholesterol level was constant [192].

For all other cholesterol precursors mevalonate-5-pyrophosphate, isopentenyl-PP, presqualene-pyrophosphate, (S)-squalen-2,3-epoxide, 4-methylzymosterol-carboxylate, 3-keto-4-methylzymosterol, 4-methylzymosterol, 4-alpha-carboxyl-5-alpha-cholesta-8,24-dien-3-beta-ol, zymosterone, 5-alpha-cholesta-7,24-dien-3-beta-ol, cholesta-8-en-3-beta-ol, 5-alpha-cholesta-7,24-dien -3-beta-ol) we could not find reports on their involvement in cancers.

We demonstrated evidence from the above literature on opposite effects of cholesterol precursors and what determines the need of further investigations to collect a more balanced view on tumor promoting and tumor suppressive roles of these metabolites.

## 5. Conclusions

Cancer significance of cholesterol synthesis pathway strengthens with the growing number of reports on transcriptomic, proteomic and metabolomic aspects. We summarized and analyzed the current data on interactomics of cholesterol synthesis enzymes in the cancer-related context, since this topic has only been fragmentarily mentioned and not sufficiently systematized. To date, PPIs and mechanisms of regulation through post-translational modifications are fairly well investigated for 3-hydroxy-3-methylglutaryl coenzyme A reductase and squalene epoxidase. Analysis of interactomics datasets showed that there is a dual problem of poor coverage of sub-interactome mapping for one part of enzymes and protein partners’ redundancy for other target enzymes. The lack of functional annotation for experimentally verified binary protein interactions, especially in the disease-related and tumor-/tissue-specific context, could stimulate investigation on sub-interactomes mapping and revealing the post-translational regulation mechanisms for understudied clinically relevant cholesterol synthesis enzymes. Future studies are also required to identify tumor/normal discrimination of protein–protein and protein–ligand interactions involved in this pathway.

According to the Cancer Dependency Map (https://depmap.org/portal/) knockouts of *HMGCR*, *MVK, MVD* and *FDPS* genes in the mevalonate “part” of the pathway resulted in a loss of cell viability of cancer cell lines. Since enzymes encoded by these genes are druggable and their pharmacological targeting can be of particular interest for development of medicines specific to cancer cells. The reasonableness of complementary cholesterol lowering strategy for increasing the efficacy of classical anticancer therapeutic schemes is being tested in several clinical trials. However, a parity of benefits and harms of pharmacological “shutdown” of discrete parts or a whole cholesterol synthesis pathway is still controversial.

## Figures and Tables

**Figure 1 biomedicines-09-00895-f001:**
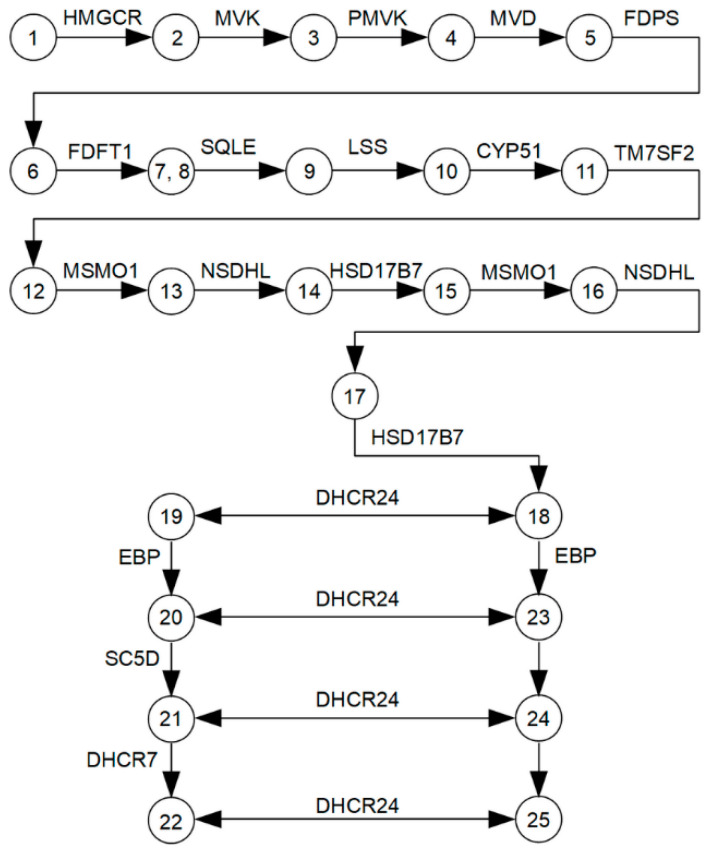
A logical scheme of enzymes and metabolites involved in the cholesterol synthesis pathway. The following metabolites are indicated by numbers: (1)—hydroxymethylglutaryl-CoA; (2)—(R)-mevalonate; (3)—(R)-5-phosphomevalonate; (4)—(R)-5-diphosphomevalonate; (5)—isopentenyl diphosphate; (6)—(2E,6E)-farnesyl diphosphate; (7)—presqualene diphosphate; (8)—squalene; (9)—(S)-squalene-2,3-epoxide; (10)—lanosterol; (11)—4,4-dimethyl-5-alpha-cholesta-8,14,24-trien-3-beta-ol (FF-MAS); (12)—14-demethyllanosterol (T-MAS); (13)—4-alpha-methyl zymosterol-4-carboxylate; (14)—3-keto-4-methylzymosterol; (15)—4-alpha-methyl zymosterol; (16)—4-alpha-carboxy-5-alpha-cholesta-8,24-dien-3-beta-ol; (17)—zymosterone; (18)—zymosterol; (19)—zymostenol; (20)—lathosterol; (21)—7-dehydrocholesterol; (22)—cholesterol; (23)—5-alpha-cholesta-7,24-dien-3-beta-ol; (24)—7-dehydrodesmosterol; (25)—desmosterol.

**Figure 2 biomedicines-09-00895-f002:**
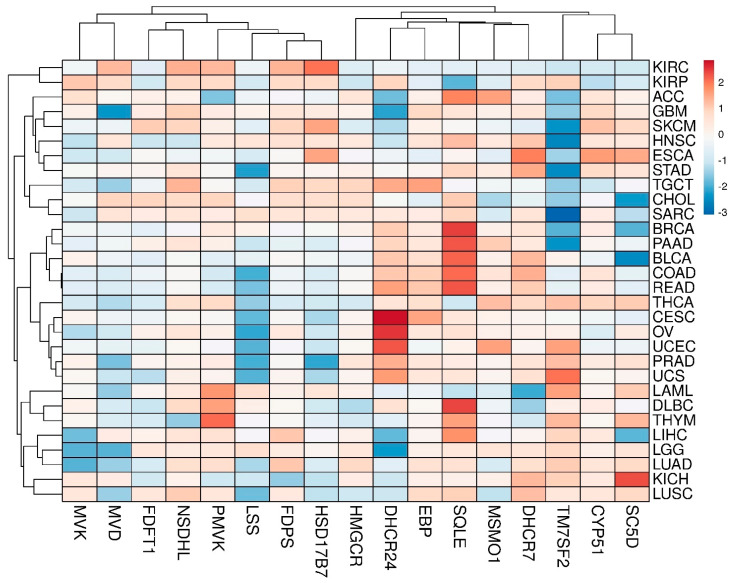
The clusterogram of expression profiles of genes, involved in cholesterol synthesis pathway, in different tumors. The color scale shows the cluster distances.

**Figure 3 biomedicines-09-00895-f003:**
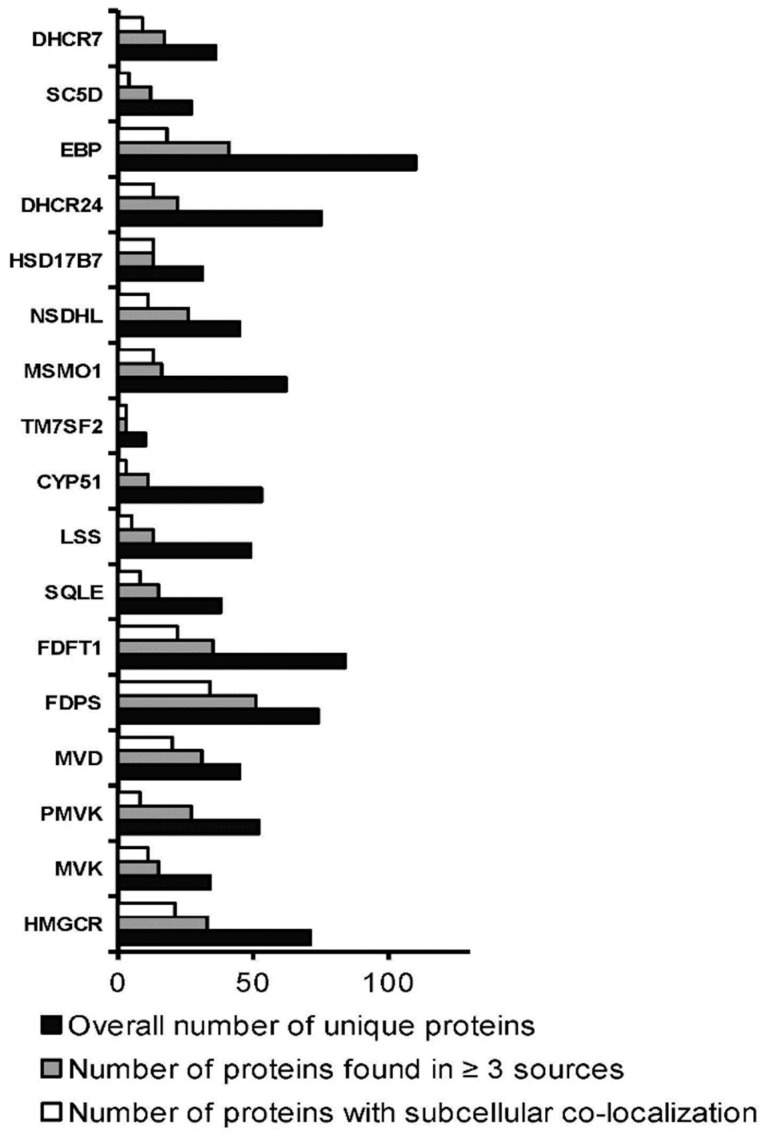
Selection rounds of found protein partners.

**Figure 4 biomedicines-09-00895-f004:**
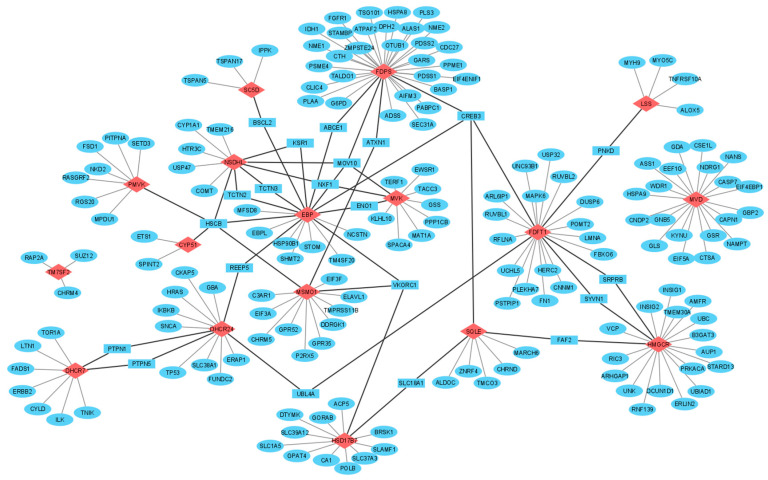
Network construction of PPIs involving the cholesterol synthesis enzymes. Legend: target enzymes, unique and common protein partners are shown in red diamonds, blue ellipses and blue rectangles, respectively. Edges, connecting enzymes through common protein partners, are highlighted in bold.

**Figure 5 biomedicines-09-00895-f005:**
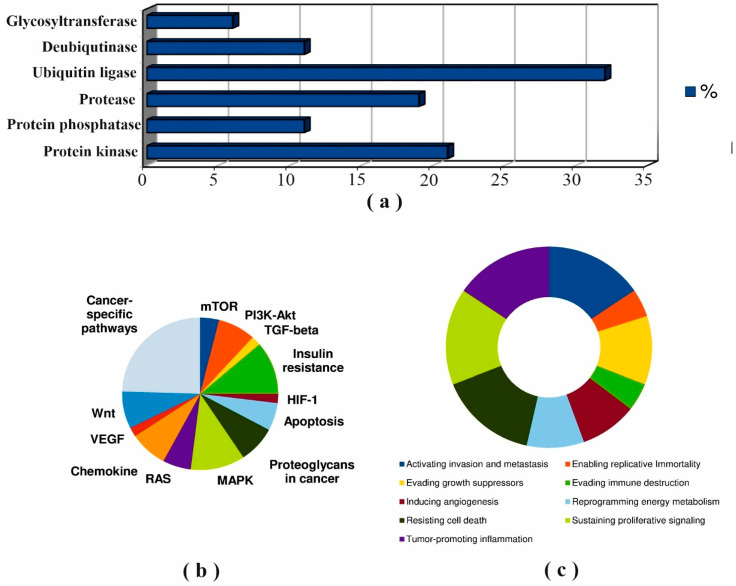
Annotation of modifying protein partners of target enzymes: distribution of functional classes (**a**); involvement in cancer pathways (**b**); enrichment with cancer hallmarks terms (**c**).

**Figure 6 biomedicines-09-00895-f006:**
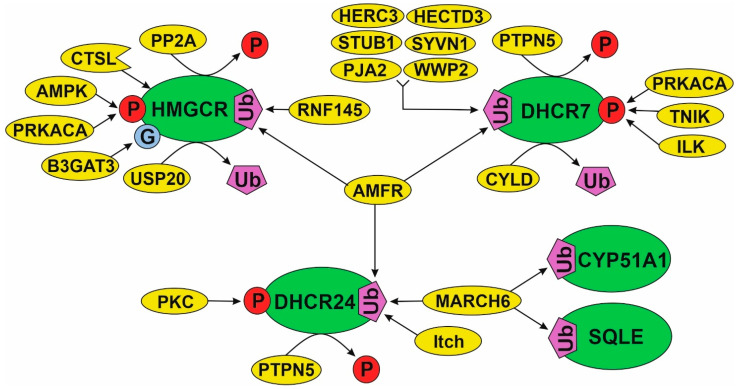
A model of the effects of modifying proteins on some cholesterol synthesis enzymes.

**Table 1 biomedicines-09-00895-t001:** Interactome browsers for retrieving protein–protein interactions data.

No.	Name and Ref.	Site	Notes
1	Funcoup 5.0 [20]	http://FunCoup.sbc.su.se (accessed on 15 March 2021)	Evidence type—protein interactions, LLR Score > 2, confidence > 0.9, network: “PPI”, “Complex”
2	Mentha [21]	https://mentha.uniroma2.it/ (accessed on 15 March 2021)	Evidence type: physical interactions
3	CORUM [22]	http://mips.helmholtz-muenchen.de/corum/ (accessed on 15 March 2021)	-
4	ExoCarta [23]	http://www.exocarta.org/ (accessed on 15 March 2021)	-
5	APID [24]	http://apid.dep.usal.es (accessed on 15 March 2021)	Methods (binary, indirect),
			≥1 publication
6	MINT [25]	https://mint.bio.uniroma2.it/ (accessed on 16 March 2021)	Interaction type: association, physical association
7	SIGNOR 2.0 [26]	https://signor.uniroma2.it/ (accessed on 16 March 2021)	-
8	HuRI [27]	http://www.interactome-atlas.org/ (accessed on 16 March 2021)	-
9	IID [28]	http://iid.ophid.utoronto.ca (accessed on 16 March 2021)	Interaction type: experimental
10	Bioplex Explorer 3.0 [29]	https://bioplex.hms.harvard.edu/explorer/ (accessed on 16 March 2021)	Interaction probability > 0.9
11	Wiki-Pi [30]	https://hagrid.dbmi.pitt.edu/wiki-pi/ (accessed on 16 March 2021)	-
12	HIPPIE 2.0 [31]	http://cbdm-01.zdv.uni-mainz.de/~mschaefer/hippie/ (accessed on 16 March 2021)	Score > 0.6
13	HINT [32]	http://hint.yulab.org/ (accessed on 16 March 2021)	Binary interaction

**Table 2 biomedicines-09-00895-t002:** Associations between cholesterol synthesis enzymes and carcinogenesis.

Enzyme	Thesis	Ref.
HMGCR	Genetically proxied inhibition of HMGCR is significantly associated with lower odds of epithelial ovarian cancer ([odds ratio 0.60 [95% CI, 0.43–0.83]).	[49]
	Inhibition of HMGCR by fluvastatin disrupts the non-small cell lung cancer (NSCLC) tumorigenesis. Knockdown of HMGCR in NSCLC cells induces apoptosis in vitro and in vivo models.	[50]
	Statin drugs, inhibiting HMGCR, increase the efficacy of some genotoxic anti-cancer drugs.	[51,52,53,54]
	HMGCR is expressed on carcinoma cells but not on normal epithelial cells in thymic tissue. Inhibition of HMGCR by fluvastatin suppresses cell proliferation and induces the carcinoma cell death.	[55]
PMVK	PMVK can be considered as a novel prognostic biomarker for high-grade serous ovarian carcinoma (HGSOC). High expression of PMVK is significantly improves the survival of patients with HGSOC (adjusted hazard ratio, 0.430; [95% CI, 0.228–0.809]).	[56]
	PMVK expression is positively correlated with drug response in estrogen receptor (ER) positive cells and negatively correlated in ER negative cells	[57]
FDPS	Increased FDPS expression is an independent risk factor of prostate cancer (PC) for early biochemical recurrence.	[58]
	FDPS inhibitors, the carnosic acid derivatives, induces apoptosis in pancreatic cancer cell lines.	[59]
	Inhibition of the FDPS in the mevalonate pathway mediates the cytotoxic effects of a platinum (II) complex with zoledronic acid against human gastric cancer cell line SGC7901.	[60]
	FDPS expression significantly correlates with TNM stage and metastasis in non-small cell lung cancer (NSCLC). Inhibition or knockdown of FDPS disrupts the TGF-β1-induced cell invasion and epithelial-mesenchymal transition (EMT).	[61]
	FDPS inhibitors improve survival of multiple myeloma (MM) patients and results in down-regulation of ERK phosphorylation in human MM cell lines.	[62]
FDFT1	FDFT1 is highly expressed in liver, lung, prostate, breast, ovary, small intestine, bladder, cervix, thyroid, and esophageal cancers. FDFT1 regulates cell cycle progression and is directly or indirectly associated with apoptotic signals.	[63]
	High expression of FDFT1 is associated with poor prognosis and promotes metastasis of lung cancer. Loss of function of FDFT1 or its knockdown significantly inhibits invasion/migration and metastasis in cell and animal models.	[64]
	FDFT1 acts as a critical tumor suppressor in colorectal cancer (CRC). Down-regulation of FDFT1 is correlated with CRC malignant progression and poor prognosis.	[65]
	The knockdown of expression or activity inhibition of FDFT1 leads to a significant decrease in prostate cancer cell proliferation.	[66]
SQLE	Overexpression of SQLE promotes lung squamous-cell carcinoma (SCC) proliferation, migration and invasion, whereas knockdown of SQLE expression shows the opposite effect. High expression of SQLE corresponds with poor prognosis in lung SCC.	[67]
	The expression of SQLE is upregulated in the hepatocellular carcinoma (HCC) tissues and its overexpression promotes cell proliferation and migration. Downregulation of SQLE inhibits the tumorigenicity of HCC cells in vitro and in vivo.	[68]
	SQLE overexpression is more prevalent in aggressive breast cancer (BC) and is an independent prognostic factor of unfavorable outcome.	[69]
	SQLE epoxidase serves as a novel prognostic biomarker for patients with HCC. Overexpression of SQLE in non-alcoholic fatty liver disease HCC tumors is significantly associated with worse overall survival and disease-free survival.	[48,70]
	SQLE reduction helps colorectal cancer cells to overcome constraints by inducing the EMT required for generation cancer stem cells. SQLE depletion disrupts the GSK3B/p53 complex, resulting in a metastatic phenotype.	[71]
	SQLE promotes nasopharyngeal carcinoma (NPC) proliferation by cholesteryl ester accumulation instead of cholesterol.	[72]
LSS	Lanosterol synthase is a molecular target for menin inhibitor leading to the loss of cholesterol homeostasis and cell death in glioma.	[73]
	LSS activity increases in the daunorubicin-resistant leukemia cell line (CEM/R2).	[74]
CYP51A1	CYP51A1 is significantly upregulated in the drug-tolerant (DT) human lung cancer cell lines. The CYP51A1 inhibitor, ketoconazole, shows the synergy in apoptosis induction with tyrosine kinase inhibitors of epidermal growth factor receptor.	[75]
	CYP51 is present at a significantly higher level in primary colorectal cancer, compared with normal colon. The strong CYP51 immunoreactivity is associated with poor prognosis.	[76]
	The VFV, a potent non-azole inhibitor of human CYP51A1, decreases the proliferation rates of lung cancer, hormone-responsive and -nonresponsive breast and skin cancer cells in a concentration-dependent manner.	[77]
TM7SF2	TM7SF2 knockout (KO) mice show no alteration in cholesterol content. However, delayed cell cycle progression to the G1/S phase was shown in TM7SF2 KO mice, resulting in reduced cell division.	[78]
	Loss of TM7SF2 increases incidence and multiplicity of skin papillomas. The null genotype shows reduced expression of nur77, a gene associated with resistance to neoplastic transformation.	[79]
NSDHL	NSDHL knockdown affects the cell cycle, survival, proliferation, and migration of breast cancer cells, resulting in suppression of breast tumor progression and metastasis. High NSDHL expression is a potential predictor of poor prognosis in breast cancer patients.	[80]
	Inhibition NSDHL can be an effective strategy against carcinomas with activated EGFR-KRAS signaling.	[81]
	The inactivation of NSDHL or its partner SC4MOL sensitizes tumor cells to EGFR inhibitors.	[82]
	NSDHL is significantly overexpressed in gastric cancer tissues that correlates with local tumor invasion, histological grade and TNM II-IV staging.	[83]
	The NSDHL up-regulated in the metastasizing mammalian mouse cell line 4T1 compared to the non-metastasizing 67NR.	[84]
DHCR24	DHCR24 knockdown reduces whereas DHCR24 overexpression enhances breast cancer stem-like cell populations, mammosphere and aldehyde dehydrogenase positive cell.	[85]
	High expression of DHCR24 in human HCC specimens correlates with poor clinical outcome. Interfering DHCR24 alters growth and migration of HCC cells.	[86]
	DHCR24 is up-regulated in bladder cancer (BC) cells compared with that in normal tissues. DHCR24 might promote the proliferation of BC cells through several cancer-associated processes.	[87]
EBP	The EBP inhibitors show the good potency and efficacy in inhibiting proliferation of human prostate cancer PC-3 cell line.	[88]
	mRNA and protein accumulation are observed in anaplastic lymphoma kinase (ALK+) tumors.	[89]
	Inhibition of the EBP leads to cancer cell death via depletion of downstream sterols.	[90]
SC5D	Decreased SC5D activity in cancer might increase prenylation of RAS, RAC or RHOC thereby promoting cancer progression.	[91]

**Table 3 biomedicines-09-00895-t003:** Gene co-expression analysis of protein–protein interactions.

Tumors	Protein–Protein Interactions
DLBC	MVK **—GSS (L—L) ***, MVK—ENO1 (L—M), MVD—EIF4EBP1 (L—M), SQLE—FAF2 (M—L), SQLE—TMCO3 (M—n/d), LSS—MYO5C (M—n/d), CYP51A1—ETS1 (n/d—H), MSMO1—C3AR1 (n/d—n/d), MSMO1—ELAVL1 (n/d—M, MSMO1—P2RX5 (n/d—M), EBP—VKORC1 (n/d—n/d), EBP—ENO1 (n/d—M)
PAAD	MSMO1—GPR35(n/d—n/d), EBP—MOV10 (L—n/d), DHCR7—FADS1 (H—M)
PRAD	DHCR24—ERAP1(n/d—M), DHCR24—CKAP5 (n/d—M), DHCR24—PTPN1 (n/d—n/d), DHCR24—REEP5 (n/d—M), DHCR24—UBL4A (n/d—M)
READ	HMGCR—VCP (M—M), DHCR24—CKAP5 (M—M), HMGCR—INSIG1 (M—n/d), DHCR7—FADS1 (H—M)
THYM	MVK—TACC3, FDPS—ATXN1, FDPS—PSME4, FDPS—TALDO1, FDPS—SEC31A, SQLE—MARCH6, SQLE—CHRND, LSS—ALOX5, LSS—MYH9, CYP51A1—SPINT2, MSMO1—ATXN1, MSMO1—ELAVL1, HSD17B7—GORAB, SC5D—IPPK, EBP—HSP90B1 (neg *), EBP—ABCE1 (neg), EBP—KSR1 (neg), EBP—TCTN2 (neg), EBP—BSCL2 (neg), EBP—MFSD8 (neg), FDPS—NME1, SQLE—FAF2, SQLE—TMCO3, MSMO1—EIF3A, EBP—NCSTN (neg)

Abbreviations: DLBC—lymphoid neoplasm diffuse large B-cell lymphoma; PAAD—pancreatic adenocarcinoma; PRAD—prostate adenocarcinoma; READ—rectum adenocarcinoma; THYM—thymoma. * Negative correlation. ** Cholesterol synthesis enzymes are underlined. *** Prevalent protein expression (according to Human Proteome Atlas): H—high; M—medium; L—low; n/d—no data.

**Table 4 biomedicines-09-00895-t004:** Cholesterol precursors in their cancer-related effects.

Metabolite	References
Group I (metabolites promoting tumors)	
Mevalonate	[50,171,172,173,174,175,176,177,178,179]
Mevalonate-5-phosphate	[180]
Squalene	[181,182]
Group II (metabolites with protective effects)	
Lanosterol	[183]
4,4 -dimethyl-cholesta-8,14,24-trienol	[184]
14-demethyl-lanosterol	[184]
7-dehydrocholesterol	[185,186,187]
Group III (metabolites with no significant effects)	
(E,E)-farnesyl-pyrophophate	[188]
Zymosterol	[189,190]
Lathosterol	[191,192]
7-dehydrodesmosterol	[191,192]
Desmosterol	[190,191,192]

## Data Availability

Not applicable.

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
