# Peer review of "Enzymes in the Cholesterol Synthesis Pathway: Interactomics in the Cancer Context"

_biomedicines, 2021, doi:10.3390/biomedicines9080895_

Round 1
Reviewer 1 Report
The manuscript is quite original and carries new data on the interactomic profiles of 17 enzymes involved in cholesterol synthesis pathway in the context of cancer development, providing novel ideas to perform further experimental investigations.
Setting The manuscript (paragraphs, sub-paragraphs...) is well organized. In addition, the tables are useful to summarize the characteristics of the cited enzymes
Language The manuscript does not require extensive language editing, except for little typos some of which have been listed below. In addition, punctuation is useful in the reading of the text
Typos:
Page 1, Introduction, line 13: “Although the number of registered clinical trials of statin therapy in cancer treatment have reached about “ It should be “Has reached”, since the subject (number) is singular. Please correct
Page 3, paragraph 3.1., lines 12,13: “may also suggested the..”, it should be “may also suggest..”. Please correct
Page 5, paragraph 3.1., line 32: “f these data concern non-human orthologues” It should be “Concerns”, since the subject (half) is singular. Please correct
Page 5, paragraph 3.1., lines 37,38: “At present, no coenzyme-dependent oxidoreductases, which can bind with target enzymes, have not yet been identified” In this phrase there is a double negative. Please correct
Page 13, paragraph 4.2.2, lines 3,4: “Loss of PGRMC1 function reduces CYP51A1 activity and increasing production of..” It should be “increases”. Please correct
Page 13, paragraph 4.2.2, lines 4,5: “..was shown to interacts..” It should be “was shown to interact” Please correct
Page 13, paragraph 4.2.4, lines 2: “There is an indication that MVD/mortalin interaction affect the..” It should be “affects”, since the subject (interaction) is singular. Please correct
Page 15, paragraph 4.3.4, lines 11: “REEP5, well as REEP6, modulating oncogenic signaling…” It should be “modulates”
Page 17, paragraph 4.5, line 9: “core anchored in the membrane that recruit other” It should be “recruits”, since the subject (core) is singular. Please correct
Page 28, paragraph 4.6, line26: “proteostasis regulation. are in Group II” there is a typo (the point), please correct
Content:
In my opinion, for the benefit of clarity, you should report the extended names of the several acronyms the first time you cite them (it could be along the text, or in figures/tables legends), and then report only the acronym. For instance, at page 2, line 9 you explain the acronym PPIs, but it was already described (page 2, line 6), so you can use it without any other explanation. And so on
Page 2, Introduction, lines 18-20: “Involvement… post-translational levels” Please add pertinent citation/citations
Page 8, paragraph 3.3, line 9 “188 unique and 18 common protein partners” in the abstract you talk about 17 common protein partners. Please correct the wrong number
Page 13, paragraph 4.2.3, lines 1,2: “It is known that Beclin 1 is an essential autophagy protein and has been shown to play a role in tumor suppression” Please insert a pertinent citation
Page 13, paragraph 4.2.4, lines 1-3: “MVD enzyme was identified as a binding partner of the mortalin which belongs to the HSP70 protein family and is involved in cell senescence and immortalization pathways”. Please insert a pertinent citation
Page 14, paragraph 4.3.1, lines 2-4: “Heat shock cognate B (HSCB) is a co-chaperone; it inserts Fe-S cluster into a number of proteins but does not have intrinsic chaperone activity and lacks a domain necessary for interaction with misfolded proteins” Please insert pertinent citation/citations.
Page 15, paragraph 4.3.5, lines 1-3: “Annotation of SC5D–BSCL2 and EBP–BSCL2 interactions is possible through the known function of BSCL2 (seipin) in the lipid droplet (LD) biogenesis. BSCL2 supports the formation of structurally uniform contacts between ER membrane and LDs” Please insert pertinent citation/citations.
Page 14, paragraph 4.3.6, lines 1-3: “The dysregulation of protein tyrosine phosphatases is associated with several human diseases, including cancers. PTPN1 directly dephosphorylates the terminal kinase c-Jun N-terminal kinase (JNK) of the MAPK pathway.” Please insert pertinent citation/citations.
Author Response
Reviewer 1
Reviewer 1 commentary:
Page 1, Introduction, line 13: “Although the number of registered clinical trials of statin therapy in cancer treatment have reached about “ It should be “Has reached”, since the subject (number) is singular. Please correct
Authors response:
Corrected accordingly.
Reviewer 1 commentary:
Page 3, paragraph 3.1., lines 12,13: “may also suggested the..”, it should be “may also suggest..”. Please correct
Authors response:
Corrected accordingly.
Reviewer 1 commentary:
Page 5, paragraph 3.1., line 32: “f these data concern non-human orthologues” It should be “Concerns”, since the subject (half) is singular. Please correct
Authors response:
Corrected accordingly.
Reviewer 1 commentary:
Page 5, paragraph 3.1., lines 37,38: “At present, no coenzyme-dependent oxidoreductases, which can bind with target enzymes, have not yet been identified” In this phrase there is a double negative. Please correct
Authors response:
The sentence was changed to the: “At present, no coenzyme-dependent oxidoreductases, which can bind with target en-zymes, have been identified”
Reviewer 1 commentary:
Page 13, paragraph 4.2.2, lines 3,4: “Loss of PGRMC1 function reduces CYP51A1 activity and increasing production of..” It should be “increases”. Please correct
Authors response:
Corrected accordingly.
Reviewer 1 commentary:
Page 13, paragraph 4.2.2, lines 4,5: “..was shown to interacts..” It should be “was shown to interact” Please correct
Authors response:
Corrected accordingly.
Reviewer 1 commentary:
Page 13, paragraph 4.2.4, lines 2: “There is an indication that MVD/mortalin interaction affect the..” It should be “affects”, since the subject (interaction) is singular. Please correct
Authors response:
Corrected accordingly.
Reviewer 1 commentary:
Page 15, paragraph 4.3.4, lines 11: “REEP5, well as REEP6, modulating oncogenic signaling…” It should be “modulates”
Authors response:
Corrected accordingly.
Reviewer 1 commentary:
Page 17, paragraph 4.5, line 9: “core anchored in the membrane that recruit other” It should be “recruits”, since the subject (core) is singular. Please correct
Authors response:
Corrected accordingly.
Reviewer 1 commentary:
Page 28, paragraph 4.6, line26: “proteostasis regulation. are in Group II” there is a typo (the point), please correct
Authors response:
The sentence was changed to the: “The anti-aggregation activity of lanosterol [https://doi.org/10.1016/j.bbamcr.2019.118617] might facilitate the cells to recycle proteins from aggregates indicating that lanosterol functions in the cancer-related proteostasis regulation.”
Reviewer 1 commentary:
In my opinion, for the benefit of clarity, you should report the extended names of the several acronyms the first time you cite them (it could be along the text, or in figures/tables legends), and then report only the acronym. For instance, at page 2, line 9 you explain the acronym PPIs, but it was already described (page 2, line 6), so you can use it without any other explanation. And so on[1]
Authors response:
Corrected accordingly.
Reviewer 1 commentary:
Page 2, Introduction, lines 18-20: “Involvement… post-translational levels” Please add pertinent citation/citations
Authors response:
The references were added accordingly.
Reviewer 1 commentary:
Page 8, paragraph 3.3, line 9 “188 unique and 18 common protein partners” in the abstract you talk about 17 common protein partners. Please correct the wrong number[2]
Authors response:
Thank you for your considerable commentary. We have checked the data and found a mistake. The number of protein partners were not calculated right. The right number of protein partners of seventeen cholesterol biosynthesis enzymes shall be 165 unique and 21 common proteins partners according to the Figure 4 and Supplementary file #3. All necessary corrections of numbers were made throughout the text.
Reviewer 1 commentary:
Page 13, paragraph 4.2.3, lines 1,2: “It is known that Beclin 1 is an essential autophagy protein and has been shown to play a role in tumor suppression” Please insert a pertinent citation
Authors response:
The reference was added accordingly.
Reviewer 1 commentary:
Page 13, paragraph 4.2.4, lines 1-3: “MVD enzyme was identified as a binding partner of the mortalin which belongs to the HSP70 protein family and is involved in cell senescence and immortalization pathways”. Please insert a pertinent citation
Authors response: The MVD (mevalonate diphosphate decarboxylase) enzyme is also named MPD (mevalonate pyrophosphate decarboxylase) (UniProtKB - P53602). The MPD-mortalin interaction is cited in this paragraph as “Wadhwa, R.; Yaguchi, T.; Hasan, M.K.; Taira, K.; Kaul, S.C. Mortalin-MPD (Mevalonate Pyrophosphate Decarboxylase) Interactions and Their Role in Control of Cellular Proliferation. Biochem Biophys Res Commun 2003, 302, 735–742, doi:10.1016/s0006-291x(03)00226-2”
Reviewer 1 commentary:
Page 14, paragraph 4.3.1, lines 2-4: “Heat shock cognate B (HSCB) is a co-chaperone; it inserts Fe-S cluster into a number of proteins but does not have intrinsic chaperone activity and lacks a domain necessary for interaction with misfolded proteins” Please insert pertinent citation/citations.
Authors response: The reference was added accordingly.
Reviewer 1 commentary:
Page 15, paragraph 4.3.5, lines 1-3: “Annotation of SC5D–BSCL2 and EBP–BSCL2 interactions is possible through the known function of BSCL2 (seipin) in the lipid droplet (LD) biogenesis. BSCL2 supports the formation of structurally uniform contacts between ER membrane and LDs” Please insert pertinent citation/citations.
Authors response: The reference was added accordingly.
Reviewer 1 commentary:
Page 14, paragraph 4.3.6, lines 1-3: “The dysregulation of protein tyrosine phosphatases is associated with several human diseases, including cancers.[3] PTPN1 directly dephosphorylates the terminal kinase c-Jun N-terminal kinase (JNK) of the MAPK pathway.” Please insert pertinent citation/citations.
Authors response:
Corrected accordingly. The JNK dephosphorylation by PTPN1 is described in the doi: 10.1038/s41598-017-13494-x. The sentence is changed to the:
The dysregulation of protein tyrosine phosphatases is associated with several human diseases, including cancers [https://doi.org/10.1038/aps.2014.80]. PTPN1 directly dephosphorylates the terminal kinase c-Jun N-terminal kinase (JNK) of the MAPK pathway; negative regulation of the JNK / MAPK pathway by PTPN1 was found to reduce the tumor necrosis factor α (TNFα)-dependent cell death response [https://doi.org/10.1038/s41598-017-13494-x]. However, it is unknown if “non-signaling” PTPN1's protein substrates exist.
Reviewer 2 Report
This is an interesting study to summarize and analyze the current data on interactomics of cholesterol synthesis enzymes in the cancer-related context since this topic has only been fragmentarily mentioned and not sufficiently systematized. Analysis of interactomics datasets showed that there is a dual problem of poor coverage of sub-interactome mapping for one part of enzymes and protein partners’ redundancy for another target enzyme. The lack of functional annotation for experimentally verified binary protein inter-actions, especially in the disease-related and tumor- / tissue-specific context, could stimulate investigation on sub-interactomes mapping and revealing the post-translational regulation mechanisms for understudied clinically relevant cholesterol synthesis enzymes. Future studies are also required to identify tumor/normal discrimination of protein-protein and protein-ligand interactions involved in this pathway. Albeit, I consider these findings to provide new insight into drug development. I still have some minor suggestions.
1, In Table3, the author analyzed the TCGA datasets and demonstrated that INSIG1 expression was markedly reduced in 19 tumors. It would be much better if the author can also validate these data on other databases, such as NCBI GEO.
2, This work is aimed at systematization and bioinformatic analysis of the available interactomics data on seventeen enzymes in the cholesterol pathway, encoded by HMGCR, MVK, PMVK, MVD, FDPS, FDFT1, SQLE, LSS, DHCR24, CYP51A1, TM7SF2, MSMO1, NSDHL, HSD17B7, EBP, SC5D, DHCR7 genes. Is this possible that the author can explore the prognosis for the above genes via Kaplan-Meier plotter ?
Author Response
Reviewer 2
Reviewer 2 commentary:
In Table3, the author analyzed the TCGA datasets and demonstrated that INSIG1 expression was markedly reduced in 19 tumors. It would be much better if the author can also validate these data on other databases, such as NCBI GEO.
Authors response:
Thank you for your very valuable comment.
In the first paragraph (subsection 4.4.1) the sentence "Analysis of TCGA datasets shows that, indeed, INSIG1 expression was markedly reduced in 19 tumors." was written contextually and logically incomplete.
We have analyzed INSIG1 gene expression in 19 pairs of TCGA normal and tumor (BLCA, BRCA, CHOL, COAD, ESCA, HNSC, KICH, KIRC, KIRP, LIHC, LUAD, LUSC, PAAD, PCPG, PRAD, READ, STAD, THCA, UCES) datasets.
CESC, SARC, SKCM and THYM datasets were discarded due to the big difference between the tumor and normal samples. It turned out that INSIG1 expression is statistically significant only in cholangiocarcinoma (CHOL) and lung adenocarcinoma (LUAD).
In accordance with the reviewer’s comment, additionally, we searched for transcriptomic datasets in the NCBI GEO for human CHOL and LUAD with statistically significant changes of INSIG1 gene expression in TCGA database. Next criteria was taken into account:
1) bulk RNA-seq of tissue samples with more than 25 cases in a cohort
2) possibility for data analysis using built-in GEO2R program for differentially expressed genes identification
3) statistical selection of gene expression by adjusted P value (Benjamini-Hochberg correction, false discovery rate) less than 0.05
4) protein-coding genes in the datasets must be annotated
We searched and analyzed GSE132305 and GSE63459, GSE43458 datasets for CHOL and LUAD, respectively.
Other found datasets GSE103909, GSE76297, GSE32225, GSE26566, GSE140797, GSE116959, GSE118370, GSE85841, GSE43767 for LUAD and CHOL as well as for READ (GSE71187) did not match the search criteria.
Table 3 contained data on the co-expression of those gene pairs for which the correlation coefficient exceeds |0.4|. The co-expression of INSIG1-HMGCR was assessed only according to TCGA datasets as a reference tumor database.
Thus, after text correction and adding the new information, the initial text fragment
“Analysis of TCGA datasets shows that, indeed, INSIG1 expression was markedly reduced in 19 tumors. However, a positive Pearson correlation (R>0.6) between up-regulated INSIG1 and HMGCR expression in case of TCGA-READ dataset.”
was replaced by the text presented below:
“Analysis of TCGA datasets shows that, indeed, INSIG1 expression was reduced in 6 from 19 tumors and was statistically significant only in CHOL and LUAD compared to normal tissues. However, under these conditions, no gene co-expression between INSIG1 and HMGCR was observed. Additional analysis of CHOL expression datasets (GSE132305, 182 tumor and 38 normal tissue samples) using NCBI-GEO depository (https://www.ncbi.nlm.nih.gov/gds) showed that INSIG1 logFC value was 0.32 (adj.P.Val = 0.026, B&H FDR). As for LUAD, we found logFC value 0.33 (adj.P.Val = 6.15*10-5) for INSIG1 expression (GSE63459 consisting of 30 paired adjacent and tumor tissue samples). In another dataset GSE43458, we compared 30 samples from non-smokers vs 40 tumor samples obtained from non-smokers and from smokers. It was shown that INSIG1 expression was characterized logFC values 0.31 (adj.P.Val = 0.01) and 0.5 (8.41*10-5), respectively. It should still be said that there was no obvious convergence between TCGA and NCBI-GEO datasets. Nevertheless, it is interesting that a positive Pearson correlation (R>0.6) between up-regulated INSIG1 and HMGCR expression in the TCGA-READ dataset (Table 3) may indicate the cancer-type specific expression of INSIG1.”
Reviewer 2 commentary:
This work is aimed at systematization and bioinformatic analysis of the available interactomics data on seventeen enzymes in the cholesterol pathway, encoded by HMGCR, MVK, PMVK, MVD, FDPS, FDFT1, SQLE, LSS, DHCR24, CYP51A1, TM7SF2, MSMO1, NSDHL, HSD17B7, EBP, SC5D, DHCR7 genes. Is this possible that the author can explore the prognosis for the above genes via Kaplan-Meier plotter ?
Authors response:
We made additional explorations and the corrected text is presented below.
The most meaningful and informative Kaplan-Meier plots and main output parameters of different variants of survival analysis are presented in Supplementary file #4.
The paragraph
“The heat map of the prognostic value (overall survival) of DEGs is shown in Figure S3 (Supplementary file #1). Kaplan-Meier survival analysis was used to select the gene signature 1 (HMGCR, DHCR7, SC5D, NSDHL, CYP51A1 and FDFT1) (Figure S4a, Supplementary file #1) and the gene signature 2 (SC5D, TM7SF2 and MVD) (Figure S4b, Supplementary file #1) for KIRC and LGG tumors, respectively. Figure S4 (Supplementary file #1) shows the decrease of overall survival by about 2.5 times in low expression groups and indicates the existence of prognostic value for both gene signatures.”
was changed to:
“The maps of overall survival (OS) and disease-free survival (RFS) for 17 target genes are shown in Figure S3a and S3b, respectively (Supplementary file #1). Five genes from these maps meet the strictest selection criteria: p<0.001, Hazard ratio (HR) ≤ 0.5 or HR ≥ 2 as well as a number of cases ≥ 200. For all of them (OS: CYP51A1 in KIRC, FDFT1 in KIRC, HMGCR in KIRC, SC5D in KIRC, TM7SF2 in LGG; RFS: FDFT1 in KIRC, SC5D in PRAD) the Kaplan-Meier plots are shown in Supplementary file #4, Figure S4-1. It should be noted that multigenic signatures have a higher priority in prognostic value than single tran-scriptome markers [https://doi.org/10.3389/fmed.2018.00248]. First of all, survival analysis of a panel of 17 target genes versus each TCGA tumor was performed (Table 1, Supplementary file #4). It is shown that the se-lection criteria were met in the case of KIRC only. More negative prognosis for survival was associated with lower profile of cholesterol biosynthesis genes expression (Figure S4-2, Supplementary File #4). It should be noted, that the severe cholesterol metabolism disrupting was observed in the KIRC cells [https://doi.org/10.1016/j.coph.2012.08.002, https://doi.org/10.1371/journal.pone.0048694, https://doi.org/10.1016/j.bbalip.2019.158525]. We also carried out the survival analysis for each target enzyme versus all tumors (pan-cancer prognostic significance). Results obtained are presented in Table 2 (Supplementary file #4), but no hits were found. Finally, the gene signature 1 (HMGCR, DHCR7, SC5D, NSDHL, CYP51A1 and FDFT1) (Figure S4a, Supplementary file #1) and the gene signature 2 (SC5D, TM7SF2 and MVD) (Figure S4b, Supplementary file #1) were found to have prognostic value for KIRC and LGG tumors, respectively. Thus, high expression of these genes can be associated with the decrease of mortality by 2.5 times (HR value = 0.4).“